# Auto Search Indexer for End-to-End Document Retrieval

**Tianchi Yang[†], Minghui Song[†], Zihan Zhang[*,†], Haizhen Huang,**
**Weiwei Deng, Feng Sun, Qi Zhang**
STCA, Microsoft Corporation
{tianchiyang,minghuisong,zihzha,hhuang,dedeng,sunfeng,zhang.qi}@microsoft.com

## Abstract

Generative retrieval, which is a new advanced paradigm for document retrieval, has recently attracted research interests, since it encodes all documents into the model and directly generates the retrieved documents. However, its power is still underutilized since it heavily relies on the "preprocessed" document identifiers (docids), thus limiting its retrieval performance and ability to retrieve new documents. In this paper, we propose a novel fully end-to-end retrieval paradigm. It can not only end-to-end learn the best docids for existing and new documents automatically via a semantic indexing module, but also perform end-to-end document retrieval via an encoder-decoder-based generative model, namely Auto Search Indexer (ASI). Besides, we design a reparameterization mechanism to combine the above two modules into a joint optimization framework. Extensive experimental results demonstrate the superiority of our model over advanced baselines on both public and industrial datasets and also verify the ability to deal with new documents.

## 1 Introduction

Search engines are widely deployed on web applications to meet users' daily information requirements (Wang et al., 2022). Given a user query, search engines usually first retrieve candidate documents from a huge document collection and then rank them to return a ranking list. Consequently, the performance and efficiency of document retrieval are essential to the final search quality.

Recently, a new end-to-end document retrieval framework named Generative Retrieval is proposed to develop a differentiable indexer, which directly maps a given query to the relevant document identifiers (docids) via a seq2seq model (Tay et al., 2022). Specifically, some policies are first applied to preprocess all the existing documents for docids such

as assigning unique integers for documents (Tay et al., 2022; Zhou et al., 2022). Given the preprocessed docids, a Transformer-based model is employed to encode the document-docid mapping information into its parameters, and meanwhile is trained to generate relevant docids directly from a given query. As such, by adding a preprocessing phase, it turns the whole index-retrieve process into a generation task.

Despite the great success of these methods, the power of generative retrieval is still underutilized since they rely on the *pre-processed* docids, thus leading to the following limitations. (1) *New documents cannot be seamlessly retrieved by an existing trained indexer.* On the one hand, docids are preprocessed so that new documents cannot obtain their docid assignments directly from the retrieval model. On the other hand, even if their docids are obtained by the same "pre-processing" policy, these new docids are usually unknown semantics to the retrieval model. (2) *Existing preprocessing policies are confined to one-to-one mapping between documents and docids.* Accordingly, only one single document can be retrieved for each retrieval calculation. It deviates from the intention of the retrieving-ranking framework, i.e., a groups of relevant documents are expected to be efficiently retrieved in the retrieving stage (Guo et al., 2022). We argue to assign similar documents with same docid, which supports retrieving more documents at the same computational cost. (3) *The preprocessing phase is independent of the index-retrieve process.* Consequently, the caused semantic gap between the docids in preprocessing phase and the embedding space in index-retrieve process limits the performance of generative retrieval. However, it is not trivial to automatically learn the best docids within a joint framework, since the docids, which is served as the generation ground-truths, cannot maintain the gradient flow because they must appear in discrete form by an argmax function. Therefore, the

---

[*] Corresponding author.
[†] The authors contribute equally.

docid learning process and the index-retrieve process are still independent of each other even if they are integrated together.

In this paper, we propose a novel fully end-to-end generative retrieval paradigm, Auto Search Indexer (ASI). It combines both the end-to-end learning of docids for existing and new documents and the end-to-end document retrieval into a generative model based joint optimization framework. Specifically, we model document retrieval problem as a seq2seq task, i.e., with user queries as input, it outputs the docids of retrieved documents. Then, we design a semantic indexing module, which learns to automatically assign docids to existing / new documents. Besides, we design two semantic-oriented losses for it, which makes semantically similar documents share the same docids and assigns different docids to dissimilar documents. As such, the new document will be assigned an existing docid based on its content, or a new docid but belonging to the same semantic space as other docids. Furthermore, a reparameterization mechanism is proposed to enable gradient to flow backward through the semantic indexing module, thus supporting joint training for all modules. Extensive experiments on public and industrial datasets show that the proposed ASI outperforms the state-of-the-art (SOTA) baselines by a significant margin in document retrieval, and demonstrate that our semantic indexing module automatically learns meaningful docids for documents.

The contributions are summarized as follows:

- To the best of our knowledge, we are the first to propose a fully end-to-end pipeline, Auto Search Indexer (ASI), which supports both end-to-end docid assigning and end-to-end document retrieval within a joint framework.

- To this end, we propose a semantic indexing module as well as two novel semantic-oriented losses to automatically assign documents with docids, and develop a reparameterization mechanism to make the individual modules optimize jointly.

- Extensive experiments demonstrate that our ASI can learn the best docid for documents, and meanwhile achieves the best document retrieval compared to the SOTA methods on both public dataset and real industrial dataset.

## 2 Related Work

Studies about document retrieval can be roughly divided into three categories: sparse retrieval, dense retrieval and generative retrieval, which are briefly introduced as follows.

### 2.1 Sparse Retrieval

Early studies are mostly based on inverted index and retrieve documents with term matching metrics, e.g., TF-IDF [45]. BM25 (Robertson and Zaragoza, 2009) measures term weights and computes relevance scores based on TF-IDF signal. Recent studies design to leverage the word embeddings to help build inverted index (Zheng and Callan, 2015; Dehghani et al., 2017; Dai and Callan, 2020b,a). To alleviate the mismatch problem between query and document words, which is the key weak point for sparse retrieval, researchers attempt to augment possible terms before building the inverted index, e.g., Doc2Query (Nogueira et al., 2019b).

### 2.2 Dense Retrieval

In another line to relieve the mismatch problem, solutions based on deep learning first embed the queries and documents to dense vectors and then retrieve documents per vector similarity (Lu et al., 2020; Ma et al., 2022; Ni et al., 2022b; Zhan et al., 2022; Li et al., 2023a; Wang et al., 2023). These methods especially benefit from recent advances in pretrained language models (PLMs). For instance, SimCSE (Gao et al., 2021) is a simple but effective contrastive learning framework that employs BERT (Devlin et al., 2019) or RoBERTa (Liu et al., 2019). To improve the inference latency, dense retrieval methods are usually equipped with approximate nearest neighbor (ANN) (Subramanya et al., 2019) or maximum inner product search (MIPS) algorithms (Shrivastava and Li, 2014) to retrieve relevant documents within a sub-linear time cost.

### 2.3 Generative Retrieval

Recently, an alternative architecture is proposed to end-to-end map user queries to relevant docids with a Transformer-based autoregressive model. Specifically, Tay et al. (2022) and Wang et al. (2022) propose to preprocess the documents into atomic identifiers or hierarchical semantic identifiers with hierarchical k-means algorithm. Differently, SEAL (Bevilacqua et al., 2022) devise to leverage all n-grams in a passage as its identifiers. Chen et al. (2022) similarly retrieves evidence by returning

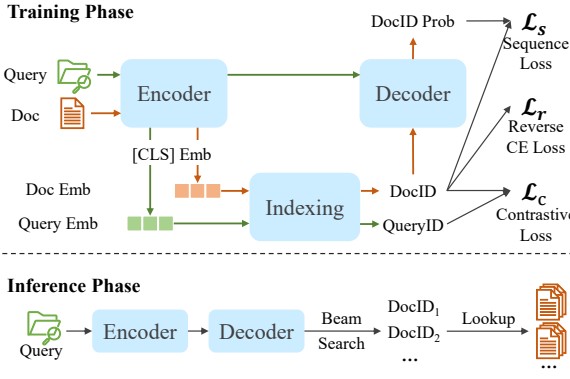

Figure 1: Illustration of the proposed model ASI.

sentence identifiers, namely GERE. Ultron (Zhou et al., 2022) designs both keyword-based identifiers and semantic-based identifiers and develops a three-stage training workflow. SE-DSI (Tang et al., 2023) proposes to use summarization texts as document identifiers. MINDER (Li et al., 2023b) assigns documents multiview identifiers. Concurrently with our work, Sun et al. (2023) also investigated a different framework, GenRet, which uses a codebook and discrete auto-encoder with progressive training to learn the doc-id assignments within retrieval stage, while it relies on a clustering-based initialization[1].

In a word, they suffer from the docid *pre-processing* phase, forming a fake end-to-end framework. In this paper, we propose a fully end-to-end paradigm ASI. It not only supports end-to-end document retrieval by a generative model, but also end-to-end learns the best docids for documents within a joint optimization framework.

## 3 Our Proposed Method

### 3.1 Overview

In this subsection, we present an overview of our novel Auto Search Indexer. The basic idea is, as illustrated in Figure 1, to build a fully end-to-end pipeline to automatically learn the meaningful docids for documents, perform end-to-end document retrieval, and combine them into a joint framework.

In detail, our ASI adopts encoder-decoder architecture to encode the user query $q$ and directly generate relevant docids $id^{(i)}, i = 1, 2, \cdots$. Distinguished from existing preprocessing-based methods, a semantic indexing module is integrated to

---

[1]Compared with ASI, GenRet additionally relies on a progressive training scheme and unique docid assignment, leading to limited training and inference efficiency for large-scale corpora.

automatically assign docids to existing / new documents. To encode semantics into the docids, we creatively design a discrete contrastive loss and a sequence-oriented reverse cross-entropy loss for the semantic indexing module, which helps to assign semantically meaningful docids and break the limitation of one-to-one mapping between documents and docids. Moreover, a reparameterization mechanism is proposed to enable gradient flowing through the indexing module to support joint optimization, thus saving the decoder from falling into meaninglessly mimicking the indexing module. In other words, the decoder thus gains the ability to surpass the indexing module on document retrieval.

### 3.2 Basic Architecture

Noticing the outstanding advances of generative PLMs, ASI adopts a seq-to-seq framework.

Specifically, ASI forms "query-to-docid" paradigm, i.e., ASI takes the user query as input and generates several relevant docids, which are represented as a sequence of id tokens. To this end, with the help of a transformer-based encoder-decoder architecture, the query $q$ is encoded by its encoder, and the generation probability is estimated by its decoder as follows,

$$\boldsymbol{h}_i = \text{Decoder}(\text{Encoder}(q), \boldsymbol{h}_{<i}), \quad (1)$$

$$P(id_i \mid q, id_{<i}; \Theta_{\text{e,d}}) = \text{Softmax}(\boldsymbol{h}_i \boldsymbol{W}), \quad (2)$$

where $id_i$ denotes the $i$-th token in the currently given docid $id$ of length $m$, which is obtained by the semantic indexing module. $\Theta_{\text{e,d}}$ denotes the trainable parameters in encoder-decoder architecture. $\boldsymbol{W} \in \mathbb{R}^{d \times V_{id}}$ is the linear parameter to classify the hidden state $\boldsymbol{h}_i$ into the docid vocabulary of size $V_{id}$. Here we treat all of the docids as different tokens from the encoder vocabulary, and therefore the encoder and decoder do not share the vocabulary space to improve decoding efficiency.

Finally, to maximize the target docid sequence likelihood, we adopt cross-entropy loss to optimize the following generation objective function,

$$\mathcal{L}_s(\Theta_{\text{e,d}}) = \sum_{(q,d) \in \mathcal{D}} \sum_i^m \log P(id|q, id_{<i}; \Theta_{\text{e,d}}), \quad (3)$$

where the target docid $id$ is obtained based on document $d$ by the semantic indexing module, which will be described next.

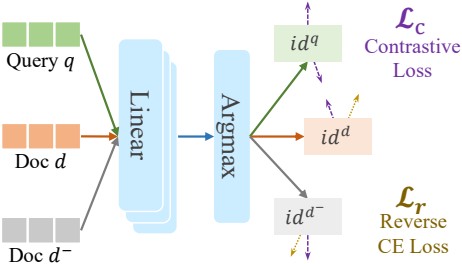

Figure 2: Illustration of the proposed semantic indexing module and two semantic-oriented losses.

## 3.3 Semantic Indexing Module

Previous works focus on the end-to-end retrieval phase, while neglecting to build an end-to-end learning framework for docid indexing, i.e., they have to "preprocess" the documents for docids (Tay et al., 2022; Wang et al., 2022; Zhou et al., 2022). Therefore, they can hardly deal with new documents, which are common and unavoidable in practical applications. To tackle the above problem, in this subsection, we propose a semantic indexing module to automatically assign docids to existing / new documents, as illustrated in Figure 2.

Specifically, given the input document $d \in \mathcal{D}$ and its representation $\boldsymbol{x}$ from encoder[2], the semantic indexing module assigns its corresponding $i$-th id token based on the following probability distribution,

$$P(id_i|d; \Theta_{e,i}) = \text{Linear}_i(\boldsymbol{x}), \quad (4)$$

$$id_i = \arg\max P(id_i|d; \Theta_{e,i}), \quad (5)$$

where $\Theta_{e,i}$ denotes the model parameter in encoder and semantic indexing module. Note that this semantic indexing module could have been more elaborately designed, while this paper focuses on the "fully" end-to-end framework, and hence this module is designed from a simple point of view.

## 3.4 Semantic-Oriented Losses

Existing indexing policies commonly enforce that each docid uniquely refers to one document, which reduces the retrieval efficiency. We argue to assign similar documents with same docid, thus supporting to retrieve more documents at the same computational cost.

To this end, as depicted in Figure 2, we propose a discrete contrastive loss and a sequence-oriented reverse cross-entropy loss for semantic indexing

[2]We represent it by its [CLS] embedding.

module to softly encourage the assignment of different docids to different documents, rather than utilizing manual rules to force it.

**Discrete Contrastive Loss** First of all, the premise to assign similar documents with same docid is that the encoder should learn semantic-based representations for documents. Accordingly, we propose a discrete contrastive loss to help learn different embeddings for documents of different semantics, where the "different" is measured by query-document pairs.

Formally, given a set of training examples $\mathcal{D} = \{(q, d)\}$ composed of query-document pairs, the discrete contrastive objective function is as follows,

$$\mathcal{L}_c = \sum_{\substack{(q,d)\in\mathcal{D} \\ (q,d^-)\notin\mathcal{D}}} \max(0, \tau(id^q, id^d) - \tau(id^q, id^{d^-}) + \alpha), \quad (6)$$

where the docid $id^d$ and pseudo query id $id^q$ are calculated by Eq.(5), $\alpha$ is a hyperparameter of margin. Note that it is hard to measure the distance between two discrete ids and calculate the corresponding gradient, thus we use the probability distribution of ids to calculate distances. Formally,

$$\tau(id^q, id^d) = \sum_i^m \|P(id_i^q|q; \Theta_{e,i}) - P(id_i^d|d; \Theta_{e,i})\|^2, \quad (7)$$

where the distribution $P(\cdot)$ refers to Eq.(4).

**Sequence-Oriented Reverse Cross-Entropy Loss** Intuitively, considering minimizing cross-entropy loss is usually used to make a variable closer to a given label, we can conversely maximize it to make an id token away from the given label (Pang et al., 2018). Considering that docid is formed as a sequence of id tokens, it is not necessary to guarantee that every token of the two sequences is different, but only one of the tokens is different. Therefore, given a pair of docids, we can find the maximum value from the cross-entropy of all id token pairs, which means this pair is most likely to become different. Then, our sequence-oriented reverse cross-entropy loss is proposed to maximize this maximum value.

Formally, given two documents $d_j, d_k \in \mathcal{D}$, let $id^j$ and $id^k$ denote their docids, respectively. We can diversify their docids by minimizing the fol-

lowing loss function,

$$\mathcal{L}_r(\Theta_{\mathrm{e,i}}) = -\sum\nolimits_{d_j \neq d_k} \mathcal{L}_r^{(j,k)}(\Theta_{\mathrm{e,i}}) , \quad (8)$$

where $\mathcal{L}_r^{(j,k)}(\Theta_{\mathrm{e,i}}) = \max_i \mathrm{CrossEntropy}\{$

$$\left(P(id_i^j \mid d_j; \Theta_{\mathrm{e,i}}), id_i^k\right)\}. \quad (9)$$

For mini-batch training, the document pairs are selected from a batch.

## 3.5 Reparameterization Mechanism

The above semantic indexing module is expected to learn the best docids jointly with $\mathcal{L}_{seq}$. However, the obtained docids are utilized as generation ground-truths, which cannot maintain the gradient flow with the help of softmax but must appear in discrete one-hot form by a gradientless argmax function. Consequently, the gradient is not able to propagate back from decoder to the semantic indexing module, thus the optimization of decoder and semantic indexing module is still decoupled. It means that directly regarding the docids of one-hot format as the training target probably makes decoder fall into meaninglessly mimicking rather than surpassing the indexing module, as $\frac{\partial \mathcal{L}_{\mathrm{seq}}}{\partial \Theta_{\mathrm{i}}} = \mathbf{0}$[3].

Inspired by DALL-E (Ramesh et al., 2021), we devise a simple but effective reparameterization mechanism to support our end-to-end learning framework via Straight-Through Estimator (STE) (Hinton, 2012). Specifically, suppose the semantic indexing module outputs the $i$-th docid $id_i$ for a document $d$, we can derive the formula according to the chain rule as follows,

$$\frac{\partial \mathcal{L}_{\mathrm{seq}}}{\partial \Theta_i} = \frac{\partial \mathcal{L}_{\mathrm{seq}}}{\partial \widehat{\boldsymbol{id}_i}} \cdot \frac{\partial \widehat{\boldsymbol{id}_i}}{\partial \mathrm{P}(\boldsymbol{id_i}|d)} \cdot \frac{\partial \mathrm{P}(\boldsymbol{id_i}|d)}{\partial \theta}, \quad (10)$$

where $\widehat{\boldsymbol{id}_i}$ denote the one-hot vector of $id_i$, and the mid-term $\frac{\partial \widehat{\boldsymbol{id}_i}}{\partial \mathrm{P}(\boldsymbol{id_i}|d)}$ is non-differential. STE suggests defining the non-differential term as "1", thus we have

$$\frac{\partial \mathcal{L}_{\mathrm{seq}}}{\partial \Theta_i} := \frac{\partial \mathcal{L}_{\mathrm{seq}}}{\partial \widehat{\boldsymbol{id}_i}} \cdot \mathbf{1}_{\arg\max \mathrm{P}(\boldsymbol{id_i}|d)} \cdot \frac{\partial \mathrm{P}(\boldsymbol{id_i}|d)}{\partial \theta}. \quad (11)$$

To this end, the outputted one-hot vector is reparameterized in forward propagation as follows,

$$\widehat{\boldsymbol{id}_i} := \widehat{\boldsymbol{id}_i} - \mathrm{detach}(\boldsymbol{P}(id_i \mid d; \Theta_{\mathrm{e,i}})) +$$
$$\boldsymbol{P}(id_i \mid d; \Theta_{\mathrm{e,i}}), \quad (12)$$

---

[3]Note that the gradient of the two semantic-oriented losses also rely on this reparameterization mechanism. Here we only take the sequence loss as an example to illustrate the reparameterization mechanism.

Table 1: Statistics of Datasets

| Datasets | ADS | MSMARCO |
|---|---|---|
| Train | 50M | 367K |
| Expansion | - | 32M |
| Valid | 10K | 5.2K |
| # Docs | 13.1M | 3.2M |

where detach() makes a tensor detached from the backpropagation. As such, STE utilizes the gradient of $\boldsymbol{P}(\boldsymbol{id}_i \mid d)$ to replace the gradient of argmax. Finally, Eq.(3) is rewritten as follows,

$$\mathcal{L}_s(\Theta_{\mathrm{e,d,i}}) = \sum_{(q,d) \in \mathcal{D}} \sum_i^m \log P(id|q, id_{<i};\Theta_{\mathrm{e,d,i}}),$$
$$(13)$$

where $\Theta_{\mathrm{e,d,i}}$ denotes the trainable parameters in encoder, decoder and indexing modules.

## 3.6 Model Training & Inference

For model training, we merge the above objective functions together as follows,

$$\mathcal{L} = \mathcal{L}_s + \gamma_c \mathcal{L}_c + \gamma_r \mathcal{L}_r, \quad (14)$$

where $\gamma_c$ and $\gamma_r$ are the scaling coefficients.

In the inference (retrieval) phase, when a user query is inputted to retrieve documents, we apply the encoder to encode the query, then adopt beam search on the decoder to generate relevant docid, and finally look up the document-docid mapping to output the retrieved documents.

As for when a new document appears and should be incorporated into the existing document collection, we apply the encoder followed by the semantic indexing module to assign a docid to it. It is worth noting that this docid belongs to the same semantic space as others. As a result, even if it has never appeared in the training document collection, there is no need to retrain the model for this docid.

# 4 Experiments

## 4.1 Experimental Setup

### 4.1.1 Datasets

We evaluate the empirical performance on a public dataset and an industrial dataset for document retrieval. The statistics of the data are reported in Table 1 and a brief introduction is as follows.

**MS MARCO Document Ranking Task** [4] (Nguyen et al., 2016) It is a large-scale dataset for machine reading comprehension, which contains a total of 3.2 million candidate documents. We use the official split of the dataset. Besides, for fair comparison, following Zhou et al. (2022), we apply DocT5Query (Nogueira et al., 2019a) for query generation.

**ADS** It is a real-world large-scale dataset collected from Bing[5] sponsored search engine, which provides organic web results in response to user queries and then supplements with sponsored ads. We collect query-ad pairs where the ads are the concatenation of the title and abstract from the sponsored ads corresponding to the user query.

### 4.1.2 Baselines

To validate the effectiveness of our proposed ASI, we compare it with the following three groups of strong document retrieval baselines.

**Sparse Retrieval** *BM25* (Robertson and Zaragoza, 2009) is a difficult-to-beat baseline which uses TF-IDF feature to measure term weights. *DocT5Query* (Nogueira et al., 2019a) utilizes T5 to generate pseudo query for document to expand document information and then applies BM25 for document retrieval.

**Dense Retrieval** We select four representative methods for comparison, i.e., *RepBERT* (Zhan et al., 2022), *Sentence-T5* (Ni et al., 2022b), *DPR* (Karpukhin et al., 2020), *SimCSE* (Gao et al., 2021) and *GTR* (Ni et al., 2022a).

**Generative Retrieval** DSI (Tay et al., 2022) is the first generative retrieval framework to directly output docids with the query as input. We compare ASI with two DSI variants, which construct docids with random unique integers and hierarchical clusters, namely *DSI-Atomic* and *DSI-Semantic*, respectively. *DSI-QG* (Zhuang et al., 2022) bridges the gap between indexing and retrieval for differentiable search index with a query generation technique. *SEAL* (Bevilacqua et al., 2022) regards all n-grams contained in documents as their identifiers. Ultron (Zhou et al., 2022) designs a three-stage training workflow where the docids are built by reversed URL or product quantization (Zhan et al., 2021) on document embeddings. We denote

these two variants as *Ultron-URL* and *Ultron-PQ*, respectively. *NCI* invents a tailored prefix-aware weight-adaptive decoder architecture, better suited to its hierarchical clustering-based docids and beam search-based generator. *GenRet* (Sun et al., 2023) uses a codebook and discrete auto-encoder with progressive training to learn the doc-id assignments within retrieval stage.

### 4.1.3 Metrics

We evaluate model performance with the following common metrics for document retrieval. Recall@K (*R@1/5/10*) treats the cases as true positives that the decoder generates the same docids as the assignments of the semantic indexing module. Moreover, for dataset ADS, Quality Score between the query and the retrieved documents is measured by an online quality estimation tool in Bing. Considering that ASI allows each docid to point to multiple documents, we report the micro- / macro-averaged Quality Score, denoted as *Mi-QS* and *Ma-QS* respectively. Besides, we also report the average number of retrieved documents for each query when generating Top10 docids, denoted as *D/Q*.

### 4.1.4 Detailed Implementation

In terms of model architecture, we build ASI with a 6-layer encoder and 6-layer decoder, where the encoder is initialized with a pretrained 6-layer BERT[6] and the decoder is optimized from scratch since its vocabulary is changed to id tokens. For encoder-only or decoder-only baselines, i.e., RepBERT, Sentence-T5, DPR, we set the layer number 12 for fair comparison. For encoder-decoder models, we set encoder/decoder layer number 6, i.e., 12 layers in total. For other settings, we set the max length of input sequence 64 for ours and follow the settings for baselines in Zhou et al. (2022). We set the margin $\alpha$=3, loss coefficients $\gamma_c$=$\gamma_r$=0.2, the length of docid $m$=4 and the range of each id token is $[0, 256)$. For model training, we set batch size 4096 and learning rate 1e-4 with AdamW optimizer. For inference, we apply vanilla beam search without constraints[7] and the beam size as 10.

---

[4]https://microsoft.github.io/msmarco/Datasets

[5]https://www.bing.com

[6]gaunernst/bert-L6-H768-uncased

[7]Note that there is no need to adopt methods such as constraint beam search to enforce that the generated docid must have corresponding documents. On the one hand, as studied in Section 4.4, new documents might occupy new docids. Consequently, the "invalid" docids have certain guiding significance for us to expand the coverage of documents. On the other hand, we have also counted the proportion of generated invalid docids, which is only about 0.58%.

Table 2: Performance on dataset MS MARCO. The best two results are shown in bold and the third best are underlined. "†" denotes that the performance is referred from Zhou et al. (2022), Sun et al. (2023) or Tang et al. (2023) and "‡" denotes we reproduce by their official implementations.

| Model | Params | R@1 | R@5 | R@10 | MRR@10 |
|---|---|---|---|---|---|
| BM25† | - | 0.1894 | 0.4282 | 0.5507 | 0.2924 |
| DocT5Query† | - | 0.2327 | 0.4938 | 0.6361 | 0.3481 |
| RepBERT† | 220M | 0.2525 | 0.5841 | 0.6918 | 0.3848 |
| Sentence-T5† | 220M | 0.2727 | 0.5891 | 0.7215 | 0.4069 |
| DPR† | 220M | 0.2908 | 0.6275 | 0.7313 | 0.4341 |
| SimCSE‡ | 110M | 0.2867 | 0.6470 | 0.7322 | 0.4390 |
| GTR-Base † | 110M | 0.4620 | - | 0.7930 | 0.5760 |
| DSI-Semantic† | 250M | 0.2574 | 0.4358 | 0.5384 | 0.3392 |
| DSI-Atomic† | 495M | 0.3247 | 0.6301 | 0.6992 | 0.4429 |
| DSI-QG † | 200M | 0.2574 | 0.4358 | 0.5384 | 0.3392 |
| NCI‡ | 376M | 0.2574 | 0.4358 | 0.5384 | 0.3392 |
| SEAL‡ | 139M | 0.2884 | 0.5683 | 0.6829 | 0.4066 |
| DynamicRetriever† | 495M | 0.2904 | 0.6422 | 0.7315 | 0.4253 |
| Ultron-URL† | 248M | 0.2957 | 0.5643 | 0.6782 | 0.4002 |
| Ultron-PQ† | 257M | 0.3155 | 0.6398 | 0.7314 | 0.4535 |
| Ultron-Atomic† | 495M | 0.3281 | 0.6490 | 0.7413 | 0.4686 |
| GenRet † | 215M | 0.4790 | - | 0.7980 | 0.5810 |
| ASI | 125M | **0.6121** | **0.7831** | **0.8207** | **0.6857** |
| ASI (Expectation) | 125M | **0.5497** | **0.7072** | **0.7414** | **0.6175** |

Table 3: Performance on dataset ADS.

| Model | R@1 | R@5 | R@10 | Mi-QS Ma-QS | D/Q |
|---|---|---|---|---|---|
| Documents in Training & Validation Set (∼13M) | | | | | |
| SEAL | 0.0444 | 0.1463 | 0.1998 | 0.5291 | 10 |
| SimCSE | 0.0934 | 0.2853 | 0.3891 | 0.3122 | 10 |
| SimCSE$_{docid}$ | 0.2768 | 0.4593 | 0.5008 | 0.5290 0.5087 | 443 |
| ASI | 0.3952 | 0.6542 | 0.7259 | 0.5053 0.4857 | 1344 |
| Full Documents Collection (∼689M) | | | | | |
| ASI | - | - | - | 0.4806 0.4661 | 142258 |

fluence of the one-to-many mapping of docid is excluded, our ASI is still significantly better than all the strong baselines. This further demonstrates the superiority of our design.

## 4.3 Retrieval Performance on ADS

Considering the sparsity of the supervision signal in datasets, the document that was not interacted with the given query according to the dataset is not definitely an "inappropriate" retrieval result. Therefore, in this subsection, we evaluate ASI on ADS dataset and focus on the retrieval quality.

**Settings.** For datasets, we first conduct experiment on the collected dataset, i.e., we train the model on training set and perform document retrieval on the about 13M documents from training & validation set. Furthermore, we also evaluate this checkpoint based on the full collection of about 689M documents in Bing platform. For baselines, we select two representative baselines, i.e., SimCSE and SEAL[8]. Furthermore, we modify SimCSE to support docid retrieval, where we use the average of document embeddings corresponding to every docid as the docid embedding, based on the document-docid mapping relationship learned by our ASI, namely SimCSE$_{docid}$.

For metrics Recall@K, as reported in Table 3, ASI outperforms the selected baselines by a greatly large margin. It does not rule out that it is because the one-to-many docids reduce the retrieval difficulty, thus we add a comparison between ASI

## 4.2 Retrieval Performance on MS MARCO

We evaluate the model performance for document retrieval on dataset MS MARCO in this subsection, which is reported in Table 2. The major findings from the results are summarized as follows:

(1) ASI significantly outperforms all the competitive baselines by a significant margin across four different metrics on the dataset MS MARCO. Especially on the R@1 metric, our ASI almost achieves twice the performance compared with the strongest baseline, i.e., Ultron-Atomic, which validates the superiority of our fully end-to-end pipeline. We attribute this surprising gain to both its tailored design and more suitable docids learned for generative retrieval.

(2) Furthermore, considering that ASI allows each docid to point to multiple documents, it is somewhat unfair for baselines to directly compare on Recall. Therefore, we modify the metrics for further comparison: For a retrieved docid that points to multiple documents, we randomly sample one single document for it. As such, ASI would also retrieve the same number of documents as baselines. Besides, in order to further eliminate the occasional performance fluctuation caused by the random sampling strategy, we choose to report the expectation of Recall/MRR in Table 2 (refer to ASI (Expectation)). It can be seen that even if the in-

---

[8]Among generative retrieval methods, SEAL relies on ngram-based docids so that new documents can be easily incorporated by simply rebuilding the FM index without retraining the model. In contrast, the others rely on virtual token-based docids that are generated during preprocessing phase, which causes difficulty in generalizing to new documents.

Table 4: Performance on new documents from ADS.

| Metrics | Full Valid | Existing | New Content | New Semantic |
|---|---|---|---|---|
| # Sample | 10000 | 1661 | 8146 | 193 |
| R@1 | 0.3952 | 0.4218 | 0.4088 | 0.0570 |
| R@5 | 0.6542 | 0.6865 | 0.6629 | 0.2073 |
| R@10 | 0.7259 | 0.7583 | 0.7333 | 0.2487 |
| Mi-QS | 0.5053 | 0.5174 | 0.5041 | 0.4640 |
| Ma-QS | 0.4857 | 0.5006 | 0.4872 | 0.4638 |
| D/Q | 1344 | 1565 | 1178 | 925 |

and SimCSE$_{docid}$. As shown in the table, there is still a remarkable improvement compared ASI with SimCSE$_{docid}$, which validates the superior performance of our ASI for docid retrieval.

For Quality Score, SEAL outperforms all the methods including ours. We analyze that this is because SEAL is based on n-grams. Limited by the sparse supervision signal in the dataset, it cannot obtain the expected Recall@K performance. But the n-grams guarantee that its retrieved documents are of high quality. After adding the docid mappings learned by ASI, SimCSE$_{docid}$ has greatly improved the Quality Score and even achieves better performance than ours, which validates the effectiveness of the docid learned by ASI in terms of quality. However, they still suffer from low retrieval efficiency. Specifically, SEAL can only generate 10 documents in one generation process (set topk=10 for beam search in decoder), and Sim-CSE also incurs high computational and storage costs even equipped with ANN Search, while our ASI could retrieve amount of documents in one generation process without any extra storage. It is worth noting that we can retrieve 1344 documents of competitive quality for each query at the same computational cost, which is more than 100 times the efficiency of SEAL.

Additionally, we apply this semantic indexing module to encode the full collection of documents, and evaluate the quality score. One can see that we can obtain more impressive retrieval efficiency and the price we need to pay is only a little bit of acceptable quality degradation. It demonstrates the advantages of our model in terms of both effectiveness and efficiency in practical usage.

### 4.4 Analysis for New Documents

As mentioned before, thanks to the semantic indexing module with two semantic-oriented losses, ASI can better deal with the new documents.

Specifically, there are two possible results when dealing with new texts. One is to assign new documents with an existing docid, which means they are new in content rather than semantics. The other is that the new documents are assigned with a new docid, which means they are new in semantics. Hence, we split the validation set into three subsets, including "Existing" denoting the documents that are contained in training set, "NewContent" denoting the first category of new documents and "NewSemantic" denoting the second category.

As shown in Table 4, compared with the performance on the full validation set of ADS, ASI performs better on group "Existing". Surprisingly, ASI also performs better in "NewContent". For the "NewSemantic" group, our ASI still achieves desirable performance. These results validate the semantic indexing module does not mechanically copy the docids contained in the training set, but fully learns the relationship between document semantics and docids. That's why our model has such a satisfying ability to handle new documents. Due to space limitations, we report the performance of baselines on new documents in Appendix A.

### 4.5 Comparison of Variants

We compare our ASI on ADS with the following variants to study the effectiveness of each module. Specifically, "ASI-Unique" denotes the variant using unique id tokens for different id positions like "0,256,512,768", which is our proposed model; "ASI-Share" denotes the variant using shared id tokens for different id positions like "0,0,0,0"; "w/o Cont" denotes the variant removing the discrete contrastive loss; "w/o RCE" denotes the variant removing the sequence-oriented reverse cross-entropy loss; "w/o Repara" denotes the variant removing the reparameterization mechanism. We also add a metric Accuracy (Acc) that counts if the semantic indexing module assigns the same docid to the query and document. As reported in Table 5, we can draw the following conclusions.

As reported in Table 5, "ASI-Unique" slightly outperforms "ASI-Share", indicating the unique id tokens for different id positions carry more semantic information. When the discrete contrastive loss is removed, the model completely fails to be trained, referring to "w/o Cont", while "w/o RCE" performs better in terms of Recall but the Quality Score drops since more documents are assigned with a same docid (i.e., the D/ID becomes larger).

Table 5: Comparisons of different variants on dataset ADS.

| Variant | R@1 | R@5 | R@10 | Mi-QS | Ma-QS | D/Q | Acc | D/ID |
|---|---|---|---|---|---|---|---|---|
| ASI-Unique | 0.3926 | 0.6701 | 0.7452 | 0.5110 | 0.4946 | 1028.3 | 0.3730 | 1.1338 |
| ASI-Share | 0.3952 | 0.6542 | 0.7259 | 0.5053 | 0.4857 | 1343.8 | 0.3944 | 1.1539 |
| w/o Cont | 0.0000 | 0.0000 | 0.0000 | 0.0000 | 0.0000 | 0.0 | 0.0000 | 10000 |
| w/o RCE | 0.4703 | 0.7239 | 0.7826 | 0.4822 | 0.4827 | 1994.7 | 0.4612 | 1.1853 |
| w/o Repara | 0.3540 | 0.6375 | 0.7092 | 0.4963 | 0.4875 | 1212.6 | 0.4075 | 1.1492 |

Table 6: Three docid cases from ADS.

| Docid 1: 245,105,149,190 (**sapphire rings**) |
|---|
| **sapphire** engagement **rings** blue nile propose ... 
 14k gold **sapphire** fine **rings** for jewelry ... 
 lab created **sapphire ring** sterling silver ... |
| Docid 2: 245,105,16,190 (**sapphire earrings**) |
| oblue **sapphire** diamond **earrings** ... 
 oblue white lab created **sapphire earrings** sterling silver ... 
 sale **sapphire** stud **earrings** fine **earrings** ... |
| Docid 3: 12,187,16,208 (**lipstick**) |
| oplum color **lipstick** target exclusions apply .. 
 onudestix **lip makeup lip products** you will love ... 
 onyx professional makeup **lipstick** ulta beauty mouse ... |

The removal of the reparameterization mechanism causes a certain decrease of both Recall and Quality. These results verify the effectiveness of each module. It is more noteworthy that the R@1 of "w/o Repara" is lower than Acc while others are not. It implies that the reparameterization mechanism exactly makes the semantic indexing module be trained jointly with encoder-decoder, so that the decoder achieves superior performance than directly using semantic indexing module.

### 4.6 Case Study for Docid Assignment

In this subsection, we provide three docid cases from ADS to uncover ASI's docid assignments.

As illustrated in Table 6, ASI can assign the same docid to semantically similar documents, e.g., documents of Docid 1 are all about sapphire rings. Besides, ASI also assigns similar docids for similar topics. For example, Docid 1 and Docid 2 differ only in the third position, so the topics are similar, i.e., sapphire rings and sapphire earrings. These cases demonstrate that our proposed model can effectively capture the document semantics and assign meaningful docids for them. Please refer to Appendix B for more detailed analysis of docid assignment, where more case analysis can be found in Appendix B.3.

## 5 Conclusion

In this paper, we make the first attempt to propose a fully end-to-end pipeline, Auto Search Indexer (ASI), which supports both end-to-end docid indexing and end-to-end document retrieval within a joint framework. Extensive experiments on the public and industrial datasets show that our ASI outperforms the SOTA methods for document retrieval. Besides, the experiments also demonstrate the superiority of ASI for handling new documents and verify the effectiveness of its docid assignments.

## Limitations

**Hierarchical Docids** The docid assigned by our ASI does not have a hierarchy due to the equivalent multiple linear layers in semantic indexing module. As studied in previous work (Tay et al., 2022; Wang et al., 2022), hierarchical docids might be more suitable for beam search-based generators. We expect this can be implemented by a hierarchical neural clustering-based indexing module. We left it for future work as we focus on the "fully" end-to-end framework in this paper.

**Docid Interpretability** The docid is represented as an integer sequence with no semantic information that humans can understand. As observed in the case study (referring to 4.6 and B.3), these integers might have become a machine language that only the model itself can understand and use during training and inference. Therefore, we will study how these integer sequences can be interpreted for humans.

**Post-Retrieval Filtering Strategy** While permitting a single docid to correspond to numerous documents may improve the efficiency of the Recall phase, it's evident that retrieving an excessive number of documents could enlarge significant pressure on the subsequent Ranking phase. This implies that a post-retrieval filtering might be required to lessen the number of retrieved results.

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

Table 7: Performance of SimCSE on new documents from ADS.

| Metrics | Full Valid | Existing | New Content | New Semantic |
|---|---|---|---|---|
| # Sample | 10000 | 1661 | 8146 | 193 |
| R@1 | 0.0934 | 0.1013 | 0.0901 | 0.0043 |
| R@5 | 0.2853 | 0.2888 | 0.2877 | 0.0130 |
| R@10 | 0.3891 | 0.3925 | 0.3884 | 0.0179 |
| QS | 0.3122 | 0.3244 | 0.3100 | 0.2987 |
| D/Q | 10 | 10 | 10 | 10 |

Table 8: Performance of SEAL on new documents from ADS.

| Metrics | Full Valid | Existing | New Content | New Semantic |
|---|---|---|---|---|
| # Sample | 10000 | 1661 | 8146 | 193 |
| R@1 | 0.0444 | 0.1295 | 0.0459 | 0.0271 |
| R@5 | 0.1463 | 0.2073 | 0.1652 | 0.0464 |
| R@10 | 0.1998 | 0.2487 | 0.2260 | 0.0656 |
| QS | 0.5291 | 0.5366 | 0.5290 | 0.5005 |
| D/Q | 10 | 10 | 10 | 10 |

## A    Baseline Performance on new documents

In this section, we report the performance on new documents of some selected baselines. Specifically, Table 7 and 8 show the performances of SimCSE and SEAL, respectively.

As shown in the above tables, the baselines similarly perform the best on group "Existing" and perform the worst in "NewSemantic". Compared these results with those in Table 4, ASI outperforms the baselines in the four groups of validation sets on the metric R@K and shows competitive performance on the metric Quality Score. More importantly, we highlight that ASI can simultaneously retrieve far more documents than the baselines with the same computational costs.

## B    Analysis on Docid Assignment

In this section, we make a thorough analysis on the docid assignment of ASI on the dataset ADS.

### B.1    Assignment Density

In ASI, each docid is allowed to point to several documents. In this section, we study the assignment density of docids.

As illustrated in Figure 3, most of the docids point to a single document, while a few of the docids point to hundreds of documents. In other words, the density of docid assignments obeys long-

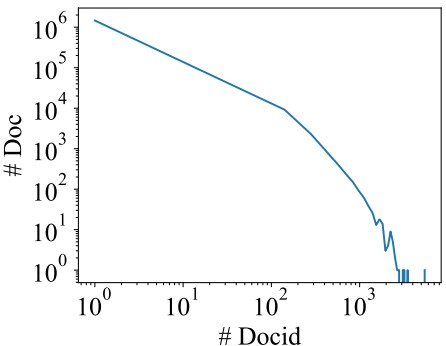

Figure 3: Assignment density of docids based on ASI.

tail distribution, which is in line with most real-world data distributions.

### B.2    Docid Visualization

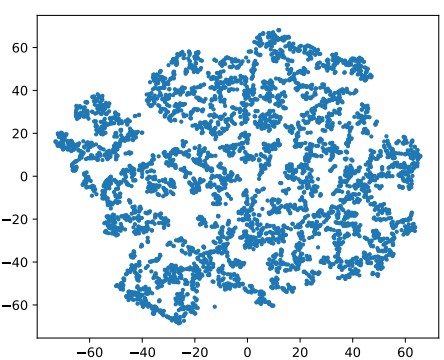

Figure 4: T-SNE visualization of docids based on ASI.

In this section, we visualize the assigned docids where the docids are regarded as continuous vector and visualized by t-SNE (Maaten and Hinton, 2008).

As depicted in Figure 4, there is no obvious clustering phenomenon on the learned docids. On the contrary, the distribution of docids is nearly uniform. It indicates that the proposed semantic indexing module as well as the two semantic-oriented losses can evenly learn from different documents. Besides, it verifies the ability of docids to distinguish different document semantics from the side and also guarantees that each docid has a large coverage of semantics.

### B.3    More Cases

We provide more docid cases from dataset ADS in Table 9, 10 and 11.

As shown in tables, it can be seen that among the four docid cases in Table 9, the first three id tokens are of the same while the fourth id token is

Table 9: Four docid cases about tablecloth from ADS.

| Docid 1: 11,200,244,50 (**green tablecloth**) |
| --- |
| **green tablecloth** bed bath beyond what product can we help you find n clearance ... 
 wayfair **green** outdoor **tablecloths** you ll love in 2023 get it by tue feb 7 ... 
 buy **green** solid **tablecloths** online at overstock our best table linens decor deals ... 
 **green** kitchen **tablecloths** bed bath beyond clearance ... 
 **green** modern elegant **tablecloths** luxury tablecloths bloomingdale's ... |

| Docid 2: 11,200,244,53 (**gingham tablecloth**) |
| --- |
| **gingham tablecloth** serena and lily over 150 new arrivals to explore the fresh start event enjoy ... 
 wayfair blue **gingham table linens** you ll love in 2023 get it by thu feb 16 ... 
 wayfair 100 cotton **gingham tablecloths** you ll love in 2023 i am in absolute love with ... 
 food network woven **gingham tablecloth** create a rustic dining atmosphere ... 
 wayfair classic farmhouse **gingham tablecloths** you ll love in 2022 n shop wayfair ... |

| Docid 3: 11,200,244,137 (**table skirt**) |
| --- |
| **table skirts** party tablecloths target exclusions apply ... 
 cloth **table skirts** wayfair ... 
 cloth **table skirts** pleated wayfair get it by thu feb 9 ... 
 9ft natural raffia **table skirt** tropical table skirts for hawaiian decoration tableclothsfactory ... 
 snap drape **table skirting** covers clips napkins more snap drape also called sdi brands was founded ... |

| Docid 4: 11,200,244,254 (**table linens**) |
| --- |
| sale tablecloths **table linens** kohl s n enjoy free shipping and easy returns every day ... 
 **table linens** tagged tablecloth page 3 elrene home fashions showing items 57 ... 
 cyber monday special tablecloth and **table linens** macy's ... 
 tablecloth and **table linens** macy's ... 
 **table linens** tagged tablecloths elrene home fashions ... |

Table 10: Five docid cases about high neck swimsuit from ADS.

| Docid 5: 10,194,75,99 (**high neck swimsuit**) |
| --- |
| **high neck swimsuit** in living art venus a feminine silhouette that offers fuller coverage ... 
 women's **high neck swimsuits** lululemon need it fast use available near you to buy and pick ... 
 waterside **high neck** one piece **swimsuit** medium bum coverage online ... 
 sailor blue **high neck** zip up **bikini** top bikini venus high neck with zipper at the ... 
 **high neck swimsuit** in living art venus r n shop high neck swim dress ... |

| Docid 6: 10,194,81,99 (**women's swim dress**) |
| --- |
| **swim dresses women**'s swimsuit dresses target exclusions apply ... 
 sale **womens swimdress** kohl s n enjoy free shipping and easy returns every day ... 
 **swim dresses women**'s clothing dillard's ... 
 **womens swim dress** lightinthebox com ... 
 **swimdress women**'s swimwear macy's new plus size cape town ... |

| Docid 7: 10,194,203,99 (**swimwear**) |
| --- |
| designer bonpoint **swimwear** saks fifth avenue n designer bonpoint swimwear at saks enjoy free shipping ... 
 best beverly **swimwear** coupon codes online stop searching start saving why scroll when you can save ... 
 ocean dream **swimwear** shop the world's largest collection of fashion shopstyle n shop 7 top ocean ... 
 ashanti **swimwear** promo codes deals discounts for free january 2023 install capital one shopping to apply ... 
 **swimwear** and beachwear zimmermann net a porter claim your exclusive discount code when you subscribe to ... |

| Docid 8: 10,194,225,99 (**bikini**) |
| --- |
| **bikini** shorts swim shorts for women venus get sexy bikini shorts that keep you on the ... 
 full coverage swim shorts **bikini** sweet dreams venus twisted bodice accentuates your curves while adding ... 
 swim shorts **bikini** aqua reef venus achieve a more modest beach day look in this pair ... 
 women's swimsuits micro **bikinis** beach cover ups asos getting ready for your holidays wherever you re ... 
 women's swimwear **bikinis** tankinis fatface us the warmest days of the year are on their way ... |

| Docid 9: 10,194,231,99 (**women's one piece swimsuits**) |
| --- |
| **women's one piece swimsuits** lululemon need it fast use available near you to buy and pick ... 
 **one piece swimsuits for women** macy's one piece swimsuits are a must have for any woman's ... 
 black **one piece women's swimsuits** swimwear macy's new women's linked in colorblocked oceanus ... 
 **one piece women's swimsuits** swimwear macy's new women's bias stripe bandeau one piece swimsuit ... 
 **one piece swimsuits for women** target exclusions apply n whether you re planning a beach vacation ... |

Table 11: More docid cases starting with "11" id token from ADS.

| Docid 10: 11,245,117,203 (**bowl**) |
| --- |
| plastic serving **bowls** williams sonoma sugg price 59 95 ...
disposable **bowls** walmart com green walmart com n shop for disposable **bowls** walmart com ...
plastic **bowls** round seagreen 2oz 100 count box my cart ... |

| Docid 11: 11,186,149,34 (**table number**) |
| --- |
| top 10 best wedding **table numbers** gold of 2023 ...
**table number** cards zazzle new instant downloads n weddings n invitations cards ...
letters **table numbers** efavormart decorations prove to be the most important aspect of a party ... |

| Docid 12: 11,170,45,92 (**marble coffee table**) |
| --- |
| pierre **marble coffee table** williams sonoma buy in monthly payments with affirm on orders over 50 ...
madison park signature **marble coffee table** with its luxurious marble top this madison park ...
buy modern contemporary **marble coffee tables** online at overstock our best living room furniture ... |

| Docid 13: 11,170,0,92 (**glass coffee table**) |
| --- |
| buy **glass** square **coffee tables** online at overstock our best living room furniture ...
st germain **glass coffee table** serena and lily buy in monthly payments with affirm on orders ...
**glass coffee tables** raymour flanigan a coffee table is for more than just coffee ... |

| Docid 14: 11,135,112,203 (**disposable plates**) |
| --- |
| **disposable plates** efavormart 5 10 off all folding chair covers ...
our 10 best **disposable plates** in the us january 2023 bestproductsreviews com ...
blue panda **disposable plates** 48 pack paper plate party supplies ... |

| Docid 15: 11,135,197,34 (**plastic tablecloth**) |
| --- |
| **plastic tablecloth** rolls in plastic tablecloths walmart com ...
free deals discounts on clear **plastic tablecloth** roll w self cutter wide thick disposable table cover ...
**plastic table cloths** etsy ... |

| Docid 16: 11,135,112,34 (**paper plates**) |
| --- |
| vdomdhtmlhtml n our 10 best **paper plates** in the us january 2023 bestproductsreviews com ...
black and white **paper plates** wayfair get it by tue feb 14 ...
**paper plates** white walmart com way to celebrate white paper dessert plates 7in 24ct ... |

| Docid 17: 11,82,244,137 (**napkins**) |
| --- |
| table linens **napkins** williams sonoma earn 10 back in rewards with the new williams sonoma credit ...
linen **napkins** when it comes to event décor every detail counts high quality banquet napkins ...
farmhouse cloth **napkins** wayfair i thought this was a great buy for that many napkins ... |

| Docid 18: 11,21,140,203 (**dinnerware**) |
| --- |
| world tableware **dinnerware** foodservice products webstaurantstore for years ...
restaurant **dinnerware** wholesale plates bowls dishes step up the elegance at your restaurant ...
tabletop **dinnerware** serveware kohl s n enjoy free shipping and easy returns every day ... |

Table 12: Quantitative similarity comparison.

|  | TF-IDF Vector | BERT Embedding |
|---|---|---|
| Random | $0.0078 \pm 0.0270$ | $0.6089 \pm 0.0820$ |
| ASI | $0.2549 \pm 0.1148$ | $0.6864 \pm 0.0469$ |
| p-value | $6.50 \times 10^{-251}$ | $5.22 \times 10^{-200}$ |

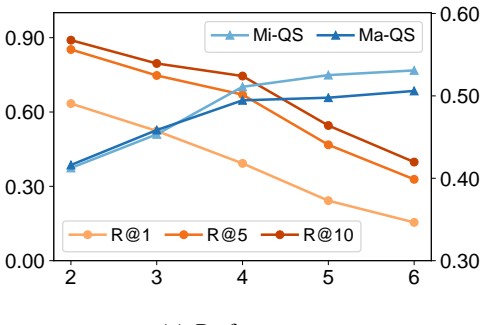

(a) Performance

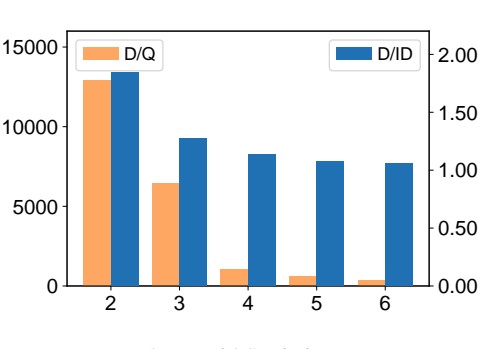

(b) Docid Statistics

Figure 5: Performance and docid statistics under different docid lengths.

different, indicating that this group of docids are similar to each other. Specifically, these four docids are all related to "tablecloths", but each has its own emphasis on details, i.e., they focus on "green tablecloth", "gingham tablecloth", "table skirt", "table linens", respectively. Similarly, the five docid cases in Table 10 are all involved "swimsuits", while they focus on the finer-grained topics "high neck swimsuit", "women's swim dress", "swimwear", "bikini" and "women's one piece swimsuits". It is worth noting that the shared id token in Table 10 appears in the first, second and fourth positions, while the different id token appears in the third position. This is because the id tokens in different positions are equivalently assigned by our semantic indexing module.

In addition, the two groups of docids in the two tables are totally different, which is also in line with the fact that the coarser-grained topics of the two are different, i.e., "tablecloths" and "swimsuits".

Moreover, we provide more cases whose docids start with "11" in Table 11. As illustrated in the table, the topics of the 9 docids are "bowl", "table number", "marble coffee tab", "glass coffee tab", "disposable plates", "plastic tableclot", "paper plate", "napkin" and "dinnerwar", respectively. Together with the cases in Table 9, these docids that start with "11" are all related to "tableware". It implies that one id token will correspond to a coarser concept, and a group of id tokens will point to a finer concept. These cases verify the effectiveness of our model to learn semantically meaningful docids for documents.

### B.4 Quantitative Analysis

In this section, we conduct a quantitative analysis for the docid assignment. In ASI, the semantic indexing mechanism allows docid to point to multiple documents based on the principle of sharing docids for similar documents and distinguishing docids for the dissimilar ones. Therefore, we measure the averaged cosine similarity of document pairs based on TFIDF vector and BERT embeddings. Specifically, "Random" means completely

random sampling, "ASI" means sampling from the same docid, and "p-value" represents whether the difference between the above two is statistically significant based on t-test (usually, $p < 0.01$ means extremely significant).

As shown in Table 12, the similarity between the documents of same docid is significantly larger than that between random documents, which demonstrates the effectiveness of docid assignments learned by ASI. We will add the above results in the revision.

## C Parameter Sensitivity

There are two essential hyper-parameters in ASI, i.e., the length of docid $m$, and the range of each id token. In this section, we study the impact of these two hyper-parameters.

### C.1 Impact of Docid Length

We evaluate the Recall@K and Quality Score for ASI of different docid lengths from 2 to 6 and the docid range is set to 256. As depicted in Figure 5(a), as the length of the docid increases, the micro- and macro-averaged Quality Scores achieve much better while the Recall@1/5/10 on the contrary decrease obviously. This is because the model capacity increases with the growth of docid length,

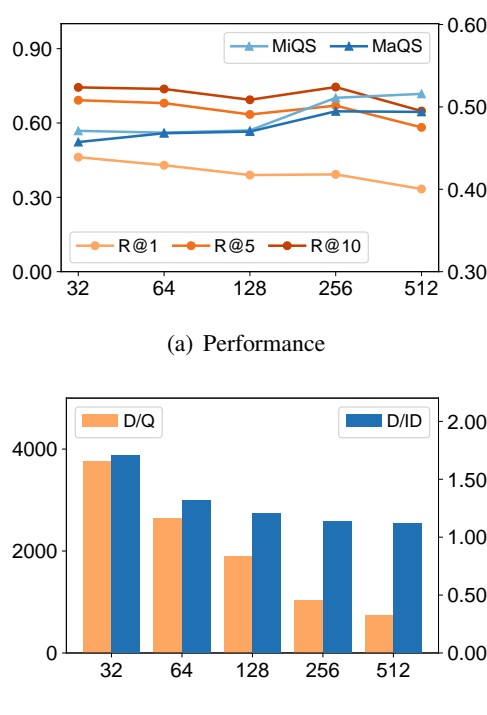

(a) Performance

(b) Docid Statistics

Figure 6: Performance and docid statistics under different docid ranges.

while the difficulty of generation for the decoder also increases due to the longer target docid. Consequently, ASI could assign different docids to documents based on their finer-grained semantics (referring to 5(b)), supporting better retrieval quality at the expense of recall.

## C.2 Impact of Docid Range

We evaluate the performance of ASI with different docid ranges from [0, 32) to [0, 512), where the length is set to 4. As illustrated in Figure 6, there are similar observations to those above for different ranges of docids. Differently, the performance of ASI is not very sensitive to the docid range. We analyze it may be due to the fact that model capacity grows exponentially with docid length, but polynomial with docid range.

All in all, the above two experimental results imply a trade-off in practical applications between efficiency and effectiveness. Therefore, in the main body of this paper, we choose to set the length of docid $m=4$ and the range of each id token is set to [0, 256).