# OpenReview forum: "Auto Search Indexer for End-to-End Document Retrieval"
_EMNLP/2023/Conference — EMNLP 2023 Findings_

### Official Review · Reviewer_WTWX · 2023-08-03

**Soundness:** 4

**Excitement:**

3: Ambivalent: It has merits (e.g., it reports state-of-the-art results, the idea is nice), but there are key weaknesses (e.g., it describes incremental work), and it can significantly benefit from another round of revision. However, I won't object to accepting it if my co-reviewers champion it.

**Paper Topic And Main Contributions:**

This paper introduces a groundbreaking approach called Auto Search Indexer (ASI), which revolutionizes the retrieval process. ASI is a novel and fully end-to-end generative retrieval paradigm that seamlessly integrates two critical components: end-to-end learning of docids for both existing and new documents, and end-to-end document retrieval.

**Reasons To Accept:**

The authors first clearly highlight the prevailing issues in generative retrieval. To alleviate these, they proposed a novel seq2seq model for generative retrieval. Their motivation makes sense and is convincing. Furthermore, the authors validate their method on two datasets, demonstrating significant improvements compared to previous works. Overall, this paper is well-structured and may make a valuable contribution to the field of generative retrieval research.

**Reasons To Reject:**

None

**Reproducibility:**

3: Could reproduce the results with some difficulty. The settings of parameters are underspecified or subjectively determined; the training/evaluation data are not widely available.

**Reviewer Confidence:**

4: Quite sure. I tried to check the important points carefully. It's unlikely, though conceivable, that I missed something that should affect my ratings.

---

> ### Author Rebuttal · Authors · 2023-08-29
>
> Thank you a lot for your positive comments!

---

### Official Review · Reviewer_agxP · 2023-08-04

**Typos Grammar Style And Presentation Improvements:** 1.	The paper is reasonably well writt…
**Soundness:** 3

**Excitement:**

4: Strong: This paper deepens the understanding of some phenomenon or lowers the barriers to an existing research direction.

**Missing References:**

1.	Sun, Weiwei, et al. "Learning to Tokenize for Generative Retrieval." arXiv preprint arXiv:2304.04171 (2023).

**Paper Topic And Main Contributions:**

This paper proposes a new end-to-end generative retrieval framework called Auto Search Indexer (ASI) for document retrieval via an encoder-decoder-based model. The main contributions are:
1.	Fully end-to-end pipeline that supports assigning and retrieval within a joint framework.
2.	Two novel semantic-oriented losses for training the indexing module.
3.	Demonstrating superior performance over existing baselines.


**Questions For The Authors:**

1.	How does the decoder ensure the validity of the output doc id?

**Reasons To Accept:**

1.	The end-to-end pipeline proposed in this paper is effective for the combination of doc-id assigning and retrieval, and resolves the new document problem.
2.	The three losses are technically sound and are shown useful for the performance.
3.	Extensive experiments have been conducted to evaluate the model on suitable datasets and metrics. And the results clearly show significant improvements over competitive baselines.


**Reasons To Reject:**

1.	Sun et al. April 2023 also proposes an end-to-end framework for learning to assigning the doc-id together with retrieval stage.
2.	More baselines on improving generative retrieval e.g., Zhuang et al., and dense retrieval e.g., Ni et al, in recent years could be included.
3.	The performance on new documents is not compared with the baseline methods.

Sun, Weiwei, et al. "Learning to Tokenize for Generative Retrieval." arXiv preprint arXiv:2304.04171 (2023).
Zhuang, Shengyao, et al. "Bridging the gap between indexing and retrieval for differentiable search index with query generation." arXiv preprint arXiv:2206.10128 (2022).
Ni, Jianmo, et al. "Large Dual Encoders Are Generalizable Retrievers." Proceedings of the 2022 Conference on Empirical Methods in Natural Language Processing. 2022.


**Reproducibility:**

3: Could reproduce the results with some difficulty. The settings of parameters are underspecified or subjectively determined; the training/evaluation data are not widely available.

**Reviewer Confidence:**

3: Pretty sure, but there's a chance I missed something. Although I have a good feel for this area in general, I did not carefully check the paper's details, e.g., the math, experimental design, or novelty.

---

> ### Author Rebuttal · Authors · 2023-08-29
>
> Thank you for your valuable comments. Below, we respond to each of your concerns in detail.
>
> $ $
>
> 1. *Regarding the related work [1] .*
>    - Thank you for providing this concurrent study. Considering it is shared in arXiv as a **preprint** within three months, we did not discuss this **concurrent work** as per the [EMNLP policy](https://2023.emnlp.org/calls/main_conference_papers/#citation-and-comparison). Although this GenRet is an end-to-end framework for learning to assign the doc-id together with retrieval stage, its structure and training processes are extremely complicated. While ours leverages two semantic-oriented losses to automatically learn the docid assignments together with retrieval in an end-to-end manner based on the Query-Doc pairs. Moreover, the performance of our ASI is significantly better than GenRet on MS MARCO (see Response 2).
>
>    - Nevertheless, we will still add the following discussion in the revision:
>
>    - Concurrently with our work, Sun et al. also investigated a totally different end-to-end framework, GenRet, which uses a codebook and discrete auto-encoder with progressive training to learn the doc-id assignments within retrieval stage. However, the decoding semantic space of GenRet (i.e., the codebook) is initialized by an isolated clustering module, while ours is learned end-to-end by ASI itself just using Query-Doc pairs, thus resulting in no semantic gap. Additionally, GenRet relies on a progressive training scheme and unique docid assignment, leading to limited training and inference efficiency for large-scale corpora.
>
> $ $
>
> 2. *Regarding the suggested baselines [2] [3].*
>
>    - Thank you for providing these SOTA baselines. We will add the comparisons as follows in the revision:
>
>      | Model                      |      Params      |         R@1          |         R@5          |         R@10         |       MRR@ 10        |
>      | :------------------------- | :--------------: | :------------------: | :------------------: | :------------------: | :------------------: |
>      | BM25 $\dagger$             |        -         |        0.1894        |        0.4282        |        0.5507        |        0.2924        |
>      | DocT5Query $\dagger$       |        -         |        0.2327        |        0.4938        |        0.6361        |        0.3481        |
>      | RepBERT $\dagger$          | $220 \mathrm{M}$ |        0.2525        |        0.5841        |        0.6918        |        0.3848        |
>      | Sentence-T5 $\dagger$      | $220 \mathrm{M}$ |        0.2727        |        0.5891        |        0.7215        |        0.4069        |
>      | DPR $\dagger$              | $220 \mathrm{M}$ |        0.2908        |        0.6275        |        0.7313        |        0.4341        |
>      | SimCSE $\ddagger$          | $110 \mathrm{M}$ |        0.2867        |        0.6470        |        0.7322        |        0.4390        |
>      | GTR-Base[3]                | $110 \mathrm{M}$ |        0.4620        |          -           |        0.7930        |        0.5760        |
>      | DSI-Semantic $\dagger$     | $250 \mathrm{M}$ |        0.2574        |        0.4358        |        0.5384        |        0.3392        |
>      | DSI-Atomic $\dagger$       | $495 \mathrm{M}$ |        0.3247        |        0.6301        |        0.6992        |        0.4429        |
>      | DSI-QG [2]                 | $200 \mathrm{M}$ |        0.2381        |          -           |        0.2805        |          -           |
>      | NCI $\ddagger$             | $376 \mathrm{M}$ |        0.2574        |        0.4358        |        0.5384        |        0.3392        |
>      | SEAL $\ddagger$            | $139 \mathrm{M}$ |        0.2884        |        0.5683        |        0.6829        |        0.4066        |
>      | DynamicRetriever $\dagger$ | $495 \mathrm{M}$ |        0.2904        |        0.6422        |        0.7315        |        0.4253        |
>      | Ultron-URL $\dagger$       | $248 \mathrm{M}$ |        0.2957        |        0.5643        |        0.6782        |        0.4002        |
>      | Ultron-PQ $\dagger$        | $257 \mathrm{M}$ |        0.3155        |        0.6398        |        0.7314        |        0.4535        |
>      | Ultron-Atomic $\dagger$    | $495 \mathrm{M}$ |        0.3281        | $\underline{0.6490}$ |        0.7413        |        0.4686        |
>      | GenRet[1]                  | $215 \mathrm{M}$ | $\underline{0.4790}$ |          -           | $\underline{0.7980}$ | $\underline{0.5810}$ |
>      | ASI                        | $125 \mathrm{M}$ |  $\mathbf{0.6121}$   |  $\mathbf{0.7831}$   |  $\mathbf{0.8207}$   |  $\mathbf{0.6857}$   |
>      | ASI (Expectation)          | $125 \mathrm{M}$ |  $\mathbf{0.5497}$   |  $\mathbf{0.7072}$   |        0.7414        |  $\mathbf{0.6175}$   |
>
>    - The performance of GTR-Base[3] and GenRet[1] are referred from [1] and DSI-QG[2]'s is referred from [4].
>
> $ $
>
> 3. *Regarding the performance of baselines on new documents.*
>
>    - Thanks for your valuable suggestions. We will add the following comparisons on new documents in the revision:
>
>      | Metrics    | Full Valid | Existing | New Content | New Semantic |
>      | :--------- | :--------: | :------: | :---------: | :----------: |
>      | # Sample   |   10000    |   1661   |    8146     |     193      |
>      |            |            |          |             |              |
>      | **ASI**    |            |          |             |              |
>      | R@1        |   0.3952   |  0.4218  |   0.4088    |    0.0570    |
>      | R@5        |   0.6542   |  0.6865  |   0.6629    |    0.2073    |
>      | R@10       |   0.7259   |  0.7583  |   0.7333    |    0.2487    |
>      | Mi-QS      |   0.5053   |  0.5174  |   0.5041    |    0.4640    |
>      | Ma-QS      |   0.4857   |  0.5006  |   0.4872    |    0.4638    |
>      | D/Q        |    1344    |   1565   |    1178     |     925      |
>      |            |            |          |             |              |
>      | **SimCSE** |            |          |             |              |
>      | R@1        |   0.0934   |  0.1013  |   0.0901    |    0.0043    |
>      | R@5        |   0.2853   |  0.2888  |   0.2877    |    0.0130    |
>      | R@10       |   0.3891   |  0.3925  |   0.3884    |    0.0179    |
>      | Mi-QS      |   0.3122   |  0.3244  |   0.3100    |    0.2987    |
>      | Ma-QS      |   0.3122   |  0.3244  |   0.3100    |    0.2987    |
>      | D/Q        |     10     |    10    |     10      |      10      |
>      |            |            |          |             |              |
>      | **SEAL**   |            |          |             |              |
>      | R@1        |   0.0444   |  0.1295  |   0.0459    |    0.0271    |
>      | R@5        |   0.1463   |  0.2073  |   0.1652    |    0.0464    |
>      | R@10       |   0.1998   |  0.2487  |   0.2260    |    0.0656    |
>      | Mi-QS      |   0.5291   |  0.5366  |   0.5290    |    0.5005    |
>      | Ma-QS      |   0.5291   |  0.5366  |   0.5290    |    0.5005    |
>      | D/Q        |     10     |    10    |     10      |      10      |
>
>    - As shown in the above table, ASI outperforms the baselines in the four groups of validation sets on the metric Recall@K and shows competitive performance on the metric Quality Score. More importantly, we highlight that ASI can simultaneously retrieve far more documents than the baselines with the same computational costs.
>
> $ $
>
> 4. Regarding how to ensure the validity of the output docid?
>
>    - Actually, there is no need to adopt methods such as constraint beam search to enforce that the generated docid must have corresponding documents. On the one hand, as studied in Section 4.4., new documents might occupy new docids. Consequently, the "invalid" docids have certain guiding significance for us to expand the coverage of documents. On the other hand, we have also counted the proportion of generated invalid docids, which is only about **0.58%**.
>
> $ $
>
> [1] Sun, Weiwei, et al. "Learning to Tokenize for Generative Retrieval." arXiv preprint arXiv:2304.04171 (2023).
>
> [2] Zhuang, Shengyao, et al. "Bridging the gap between indexing and retrieval for differentiable search index with query generation." In The First Workshop on Generative Information Retrieval at SIGIR2023. 2023.
>
> [3] Ni, Jianmo, et al. "Large Dual Encoders Are Generalizable Retrievers." In Proceedings of the 2022 Conference on Empirical Methods in Natural Language Processing. 2022.
>
> [4] Tang, Yubao, et al. “Semantic-Enhanced Differentiable Search Index Inspired by Learning Strategies.” In Proceedings of the 29th ACM SIGKDD Conference on Knowledge Discovery & Data Mining. 2023.

---

### Official Review · Reviewer_rkeY · 2023-08-07

**Soundness:** 2

**Excitement:**

3: Ambivalent: It has merits (e.g., it reports state-of-the-art results, the idea is nice), but there are key weaknesses (e.g., it describes incremental work), and it can significantly benefit from another round of revision. However, I won't object to accepting it if my co-reviewers champion it.

**Paper Topic And Main Contributions:**

This paper proposes an interesting idea to generate document ids based on document semantics for generative retrieval. With this method, generative retrieval model can be easily applied to new documents.

**Questions For The Authors:**

- For multiple documents share the same doc id, have the authors manually check the similarity of those documents?

**Reasons To Accept:**

- Expanding generative retrieval model to new documents is an important research problem.

- It is an interesting idea to generate document ids based on the semantics of document content.

**Reasons To Reject:**

- The major concern with this paper is the performance of the proposed model.  Since one doc id could be matched to multiple documents, the recall/MRR metric computation is unfair for baselines. Though the authors compute the expectation of Recall/MRR, essentially the proposed model still look for documents at more positions than the baselines.

- Some key technical details are missing. When one doc id is associated with multiple documents, how those documents will be ranked? Are they considered equally relevant to the query?

- The generative retrieval baseline in ADS dataset is weak. From the experiment in MS MARCO dataset, SEAL is not the best performing generative retrieval baseline. The authors should report the results for stronger baseline like the Ultron-Atomic.

**Reproducibility:**

2: Would be hard pressed to reproduce the results. The contribution depends on data that are simply not available outside the author's institution or consortium; not enough details are provided.

**Reviewer Confidence:**

4: Quite sure. I tried to check the important points carefully. It's unlikely, though conceivable, that I missed something that should affect my ratings.

---

> ### Author Rebuttal · Authors · 2023-08-29
>
> Thank you for your valuable comments. Below, we respond to each of your concerns in detail.
>
> $ $
>
> 1. *Regarding the comparison of recall/MRR metrics when one docid matches multiple documents.*
>
>    - Yes, when one docid points to multiple documents, it is indeed somewhat unfair for baselines to directly compare on Recall/MRR. To tackle this issue, as described on Line 488-495, we have **randomly sampled a single document for each of those docids**. Under this simple strategy, our ASI would also **retrieve the same number of documents as baselines** without additional prior knowledge, which means that our ASI did **NOT look for documents at more positions** than the baselines. Besides, in order to further eliminate the occasional performance fluctuation caused by the random sampling strategy, we choose to report the expectation under this random strategy, i.e., ASI(Expectation) in Table 2. As such, a fairer comparison is finally achieved and the results demonstrate that when retrieving the same number of documents, ASI still outperforms the baselines. We will clarify this point in the revision.
>
> $ $
>
> 2. *Regarding the document ranking strategy if one docid is associated with multiple documents.*
>
>    - This is a valuable research question. In this paper, we do not employ any ranking strategy, but simply consider all the retrieved documents equally relevant to the query. Specifically, in the evaluation of ASI(Expectation), we simply apply **random** sampling strategy. For the calculation of MRR metric, the documents of the same docid are randomly sorted, and the documents of different docids are arranged in groups according to the order of docid. This is because we focus on the "fully" end-to-end framework in this paper, and thus we leave the document ranking strategy for future work.
>
>    - PS: As expected, an effective ranking strategy can greatly benefit the one-to-many id mapping of ASI. Another experimental result on our real applications shows that the relevance score is improved nearly 3 times (0.0576 $\rightarrow$ 0.1602) when we adopt an online lightweight ranking model as a post-processing filter.
>
> $ $
>
> 3. *Regarding the generative retrieval baseline in ADS dataset.*
>
>    - In fact, we have already conducted extensive experiments on MS MARCO dataset, which provides sufficient comparisons for the generative retrieval baselines.
>
>    - Furthermore, considering new documents will frequently appear in practical applications, it is necessary for the retrieval model to be able to handle new documents. Accordingly, on dataset ADS, we aim at a more comprehensive evaluation system such as the retrieval quality (quality score) and the performance on new documents (Section 4.4). To this end, we select SEAL as the generative retrieval baseline on ADS dataset, which relies on ngram-based docids so that new documents can be easily incorporated by simply rebuilding the FM index without retraining the model. In contrast, the other methods (NCI, DSI, DynamicRetriever, Ultron-Atomic, etc.) rely on virtual token-based docids that are generated during preprocessing phase, which makes it difficult for the model to generalize to new documents. We will clarify this point in the revision.
>
>    - As an improper alternative, we preprocess these new documents from validation set *directly* with the training set, although this setting actually cannot be applied in practice as the new documents are not predictable. Here we report the performance of Ultron-PQ under this cheat setting (Ultron-Atomic will cause OOM due to the huge scale of document corpora):
>
>       |         Model         |  R@ 1  |  R@5   |  R@10  | Mi-QS  | Ma-QS  | D/Q  |
>       | :-------------------: | :----: | :----: | :----: | :----: | :----: | :--: |
>       |         SEAL          | 0.0444 | 0.1463 | 0.1998 | 0.5291 | 0.5291 |  10  |
>       |        SimCSE         | 0.0934 | 0.2854 | 0.3891 | 0.3122 | 0.3122 |  10  |
>       | SimCSE$_\text{docid}$ | 0.2768 | 0.4593 | 0.5008 | 0.5290 | 0.5087 | 443  |
>       |       Ultron-PQ       | 0.0281 | 0.0720 | 0.0854 | 0.4343 | 0.4302 |  10  |
>       |          ASI          | 0.3952 | 0.6542 | 0.7259 | 0.5053 | 0.4857 | 1344 |
>
>    - As shown in the above table, even though Ultron obtains the content of the new documents via cheat settings, it still does not outperform, which proves the superiority of ASI. Besides, we find its performance on ADS is far worse than that on MSMARCO. We analyze that Ultron is based on non-semantic virtual tokens, as also studied in [1], which makes it difficult to handle large-scale corpora.
>
> $ $
>
> 4. *Regarding the manual check of document similarity of same docid.*
>
>    - Yes, we have checked. Consequently, as shown in Section 4.6 and Appendix A.3, we have provided several cases to qualitatively uncover ASI's docid assignments. One can see that the documents of the same docid are exactly similar. From a quantitative point of view, the quality score on the ADS dataset in Table 3 can also reflect the similarity of documents of the same docid, since better quality means that the document semantics of the same docid are all relevant to the given query).
>
>    - In addition, we provide a more direct quantitative comparison here: the averaged cosine similarity of document pairs based on TFIDF vector and BERT embeddings. "Random" means completely random sampling, "ASI" means sampling from the same docid, and "p-value" represents whether the difference between above two is statistically significant based on t-test (usually, $p<0.01$ means extremely significant).
>
>       |         | TF-IDF Vector           | BERT Embedding          |
>       | ------- | ----------------------- | ----------------------- |
>       | Random  | 0.007862 $\pm$ 0.027043 | 0.608963 $\pm$ 0.082007 |
>       | ASI     | 0.254954 $\pm$ 0.114874 | 0.686494 $\pm$ 0.046917 |
>       | p-value | $6.50\times 10^{-251}$  | $5.22\times 10^{-200}$  |
>
>    - As shown in the above table, the similarity between the documents of same docid is significantly larger than that between random documents, which demonstrates the effectiveness of docid assignments learned by ASI. We will add the above results in the revision.
>
>
>
>  [1] Pradeep, Ronak, et al. "How Does Generative Retrieval Scale to Millions of Passages?." arXiv preprint arXiv:2305.11841 (2023).

---

### Official Review · Reviewer_CdLZ · 2023-08-12

**Soundness:** 4

**Excitement:**

4: Strong: This paper deepens the understanding of some phenomenon or lowers the barriers to an existing research direction.

**Paper Topic And Main Contributions:**

This paper presents an end-to-end generative information retrieval pipeline, Auto Search Indexer (ASI), that supports document-id assignment as well as document retrieval. ASI follows a seq-2-seq architecture that encodes user query q and generates the relevant documents. The proposed approach is novel since the previous work relies on preprocessed doc_ids and trained indexers can not be utilized with new documents. The model is trained with respect to a sequence loss and two semantic-oriented losses. The semantic-oriented losses are 1) contrastive loss between the q_id and pos_doc_id and 2) Reverse Cross Entropy loss between the pos_doc_id and neg_doc_id with the intuition of making the two id token distant from each other. The authors perform extensive experiments on one public and one industrial dataset and the results show solid improvement over the baseline.

**Reasons To Accept:**

The paper is well-written and well-structured. The proposed model is novel and addresses one of the fundamental challenges of the new paradigm of generative information retrieval. The baselines are thorough and the results show substantial improvement over all the baseline models including sparse retrieval, dense retrieval, and generative retrieval.

**Reasons To Reject:**

N/A

**Reproducibility:**

4: Could mostly reproduce the results, but there may be some variation because of sample variance or minor variations in their interpretation of the protocol or method.

**Reviewer Confidence:**

3: Pretty sure, but there's a chance I missed something. Although I have a good feel for this area in general, I did not carefully check the paper's details, e.g., the math, experimental design, or novelty.

---

> ### Author Rebuttal · Authors · 2023-08-29
>
> Thank you a lot for your positive comments!

---

### Meta-Review · Area_Chair_ybWy · 2023-09-23

**Recommendation:** 3

**Metareview:**

This submission studies the generative retrieval and focuses on the problem of document indexing to handle new documents properly. In particular, this submission proposes a new framework with a reparameterization mechanism to combine two modules of semantic indexing and generation. Extensive experimental results on both public and industrial datasets validate the effectiveness of the proposed method. Active discussion and additional experiments addressed most concerns from reviewers and further demonstrated the rationality of the proposed method.

---

### Decision · Program_Chairs · 2023-10-07

**Decision:**

Accept-Findings

**Comment:**

This submission studies the generative retrieval and focuses on the problem of document indexing to handle new documents properly. In particular, this submission proposes a new framework with a reparameterization mechanism to combine two modules of semantic indexing and generation. Extensive experimental results on both public and industrial datasets validate the effectiveness of the proposed method. Active discussion and additional experiments addressed most concerns from reviewers and further demonstrated the rationality of the proposed method.